# Radiotherapy and Systemic Treatment for Leptomeningeal Disease

**DOI:** 10.3390/biomedicines12081792

**Published:** 2024-08-07

**Authors:** Kelsey M. Frechette, William G. Breen, Paul D. Brown, Ugur T. Sener, Lauren M. Webb, David M. Routman, Nadia N. Laack, Anita Mahajan, Eric J. Lehrer

**Affiliations:** 1Department of Radiation Oncology, Mayo Clinic, Rochester, MN 55905, USA; breen.william@mayo.edu (W.G.B.); brown.paul@mayo.edu (P.D.B.); routman.david@mayo.edu (D.M.R.); laack.nadia@mayo.edu (N.N.L.); mahajan.anita@mayo.edu (A.M.); lehrer.eric@mayo.edu (E.J.L.); 2Department of Neuro-Oncology, Mayo Clinic, Rochester, MN 55905, USA; sener.ugur@mayo.edu (U.T.S.); webb.lauren@mayo.edu (L.M.W.)

**Keywords:** leptomeningeal disease, LMD, radiotherapy, craniospinal irradiation, immune checkpoint inhibitor

## Abstract

Leptomeningeal disease (LMD) is a devastating sequelae of metastatic spread that affects approximately 5% of cancer patients. The incidence of LMD is increasing due to advancements in systemic therapy and enhanced detection methods. The purpose of this review is to provide a detailed overview of the evidence in the detection, prognostication, and treatment of LMD. A comprehensive literature search of PUBMED was conducted to identify articles reporting on LMD including existing data and ongoing clinical trials. We found a wide array of treatment options available for LMD including chemotherapy, targeted agents, and immunotherapy as well as several choices for radiotherapy including whole brain radiotherapy (WBRT), stereotactic radiosurgery (SRS), and craniospinal irradiation (CSI). Despite treatment, the prognosis for patients with LMD is dismal, typically 2–4 months on average. Novel therapies and combination approaches are actively under investigation with the aim of improving outcomes and quality of life for patients with LMD. Recent prospective data on the use of proton CSI for patients with LMD have demonstrated its potential survival benefit with follow-up investigations underway. There is a need for validated metrics to predict prognosis and improve patient selection for patients with LMD in order to optimize treatment approaches.

## 1. Introduction

### 1.1. What Is Leptomeningeal Disease

Leptomeningeal disease (LMD) is the dissemination of malignant cells into the leptomeninges (arachnoid membrane and pia mater) or cerebrospinal fluid (CSF) that surrounds the brain and spinal cord. LMD most commonly arises from solid tumors that have a propensity to spread to the central nervous system, such as breast cancer, lung cancer, and melanoma. Other contributors to the development of LMD include surgical seeding of the meninges or tumor spillage into the CSF after the resection of brain tumors [1,2,3]. The prevalence of LMD is estimated to be 5% in patients with metastatic cancer [4,5]. The incidence of LMD may be on the rise due to advancements in systemic therapies and improved patient survival, increased detection with routine imaging and sophisticated CSF analyses, as well as changes in radiation therapy practices and recognition of nodular sub-types of LMD [6,7,8,9]. Autopsy studies have suggested that the subclinical rate of LMD may be as high as 30% [10].

### 1.2. Methods

A literature search of PubMed/MEDLINE, the Cochrane Library, and Embase was conducted to identify relevant articles reporting on LMD. Pertinent articles including the epidemiology, diagnosis, symptoms, treatments, prognosis, and active clinical trials related to LMD were included for review. Articles printed in English published between 1960 and 2024 were included in the search. Keywords used to search for relevant articles included “leptomeningeal disease”, “LMD”, “radiotherapy”, “craniospinal irradiation”, and “metastasis”. Articles discussing benign tumors were excluded from review.

### 1.3. Increasing Incidence and Recognition of LMD

The widespread utilization of targeted therapies and immunotherapy has significantly improved patient survival, including patients with brain metastases or secondary CNS disease [11]. For example, patients treated initially with human epidermal growth factor receptor 2 (HER2) therapy were noted to have a paradoxically high rate of brain metastases by unmasking CNS disease as a sanctuary site and providing improved cancer control elsewhere [12]. Immunotherapy has also improved survival including for patients with brain metastases. The Checkmate 227 study showed that dual immune checkpoint inhibitor (ICI) therapy offers a survival benefit over chemotherapy alone for stage IV non-small cell lung cancer [13]. The combined ICI approach appears to be beneficial for advanced melanoma, as ipilimumab plus nivolumab provide the longest progression-free and OS compared with ICI monotherapy [14]. Overall, improved survival and limited CNS penetrance may explain a significant proportion of patients noted to have LMD, although increased detection also likely contributes.

The decreased use of whole brain radiotherapy (WBRT) in favor of focal stereotactic radiosurgery (SRS) for brain metastases has also led to an increased risk of LMD [15,16,17]. In particular, the sub-type associated with postoperative SRS is nodular LMD. This is thought to be a distinct pattern of LMD associated with fewer symptoms and better survival [15]. The risk of LMD after postoperative SRS for brain metastases is approximately 10–15% [18]. Preoperative SRS, however, may decrease the risk of LMD after resection for brain metastases [19]. This type of nodular LMD as compared to classical diffuse sub-type LMD is displayed in (Figure 1). The latter has a characteristic linear enhancement or “sugar coating” along the leptomeninges and/or cranial nerves seen on MRI. Nodular sub-type LMD is a focal region of enhancement associated with postsurgical development as described above [15]. Nodular LMD can be difficult to distinguish from brain metastases, making the diagnosis of LMD likely previously under-reported but more commonly recognized today [20]. Prognosis for LMD has historically been poor with a median survival [2,3,4] of months despite treatment with either palliative radiotherapy or chemotherapy [21,22,23,24]. Nodular LMD is associated with a better prognosis, with a median overall survival (OS) of 8.2 months vs. 3.3 months for diffuse LMD15.

With the widespread availability of imaging modalities, such as magnetic resonance imaging (MRI), which has a 76% sensitivity and 77% specificity, more patients are undergoing cross-sectional imaging, which is more sensitive for detecting LMD [25]. CSF cytology remains the gold standard for LMD diagnosis, with a reported 100% specificity; however, the sensitivity is approximately 75%. Sensitivity is improved with repeated sampling, but this is cumbersome for patients [26]. As a result, negative CSF studies can be challenging to interpret. Therefore, MRI is particularly valuable in this setting, as contrast enhancement of the leptomeninges in the brain, cranial nerves, spinal cord, or spinal nerve roots can support the diagnosis of LMD.

Due to the limitations of conventional cytology, there has been interest in alternative CSF diagnostics such as CSF circulating tumor cells (CTCs) and CSF cell-free tumor DNA (ctDNA). Commercially available assays are available for the detection of CTCs in CSF, which can detect tumor cells based on cell surface proteins such as an epithelial cell adhesion molecule (EpCAM), which is a transmembrane glycoprotein detected in cells of epithelial origin [27]. Flow cytometry can also be used to identify CTCs in CSF based on fluorescently labeled antibodies against proteins on the tumor cell surface [28]. CSF CTC analysis permits the enumeration of tumor cells within the CSF compartment, potentially representing a biomarker that can be monitored to assess disease response [26,29]. The limitations of this technology include limited availability, the need for specific assays to detect CTCs based on primary tumor type, and potential for epithelial-to-mesenchymal transition preventing CTC detection. In contrast, ctDNA analysis is based on DNA fragments shed from tumor cells. Studies have previously established the feasibility of detecting ctDNA in a number of malignancies such as oropharynx and lung cancer [30,31]. Current investigations are assessing the utility of ctDNA for the prediction of oncologic outcomes as well as in guiding treatment decisions. In the setting of LMD, there is a growing body of evidence that ctDNA can be detected via next-generation sequencing (NGS) and can provide higher sensitivity than CSF studies or MRI findings [32]. The FORSEE trial is a randomized prospective trial monitoring ctDNA in the CSF of LMD patients to assess treatment response that recently closed to accrual in the U.S. (NCT05414123).

### 1.4. Prognosis and Optimal Treatment Selection

As noted earlier, LMD is associated with a dismal prognosis. The median survival across all patients with LMD is approximately 2–4 months with treatment, while the median survival without treatment is much shorter, approximately 4–6 weeks [33]. Variability in prognosis exists and depends on cancer sub-type, hormone receptor status, and mutational status [34]. LMD secondary to breast cancer reportedly has a better prognosis compared to LMD from lung cancer [35,36]. HER2-positive and hormone receptor-positive breast cancer confers a better prognosis compared to hormone receptor-negative breast cancer [37]. Epidermal-like growth factor (EGFR) mutated lung cancer has prolonged survival for patients with LMD when using targeted therapy [38].

The European Association of Neuro-Oncology (EANO) and European Society of Medical Oncology (ESMO) guidelines are available for the management of LMD based on the tumor type, presence of brain metastases, and extent of systemic disease [39,40]. However, there are no known curative options for LMD related to solid tumors at this time; therefore, treatment focus has been largely in the palliative setting. Additionally, the majority of patients with LMD will have neurologic symptoms that can have a significant impact on quality of life. Treatments for LMD that maximize symptom control while minimizing additional toxicity should be at the forefront of consideration for these patients. For patients who have minimal systemic disease, good performance status, and no major neurologic deficits, treatment options according to the National Comprehensive Cancer Network guidelines include central nervous system (CNS)-penetrating systemic therapies or intrathecal chemotherapy (ITC) as well as a range of radiotherapy options including focal radiotherapy, WBRT, and CSI based on clinical context [41].

A wide range of treatment options are available including focal and systemic therapies, each of which have unique toxicity profiles. Given that there is a suggestion of a spectrum of survival for LMD based on sub-type, there is room for individualized treatment. Previously identified predictors of better survival include lack of bulky disease, low CSF protein levels, lack of neurological deficits, younger age, and controlled systemic disease [42,43]. A study by Hyun et al. that reviewed 519 patients with LMD also found that patients with a better performance status (KPS ≥ 70 and normal CSF protein) had more favorable prognoses with treatment [44]. Unfortunately, currently there are no validated metrics for predicting outcomes. A group from Germany suggested a simple prognostic scoring system for patients who may benefit from certain therapies would be useful; however, such a scoring system does not exist [45]. Currently, patients are not well selected for specific treatment options. Patient selection can be challenging in general but is especially important to determine ideal therapies for optimizing outcomes and quality of life for patients with LMD.

## 2. Treatment Options for LMD

### 2.1. Wide Array of Radiotherapy Options for LMD

Radiotherapy is commonly used for the treatment of LMD. Advantages include that it is non-invasive and able to penetrate through the blood–brain barrier, which is of concern with certain systemic therapies. Radiotherapy techniques continue to evolve and allow for more conformal treatments, which can reduce toxicity as well as allow for potential dose escalation. There are a number of radiotherapy options that are available for the treatment of LMD including whole brain radiotherapy (WBRT), stereotactic radiosurgery (SRS), craniospinal irradiation (CSI), and more.

### 2.2. Whole Brain Radiotherapy

For decades, WBRT has been commonly used for LMD. In general, WBRT is well tolerated in this context [46]. WBRT has been shown to decrease neurologic symptoms and improve quality of life in select patients with LMD [47]. Furthermore, there is evidence that it is most effective at improving symptoms such as nausea and headaches with less effectiveness seen for somnolence and seizures [46]. Despite being a common treatment choice, there is not a demonstrated survival advantage with the use of WBRT for LMD [48]. In addition, there is strong evidence that WBRT can lead to neurocognitive decline as early as weeks after treatment [49,50,51]. These neurocognitive changes can impact quality of life, which is a concern for patients with a limited lifespan. Clinicians commonly continue to utilize WBRT when there is diffuse LMD or bulky disease or patients have a poor performance status [48].

### 2.3. Focal Radiotherapy—Stereotactic Radiosurgery and External Beam Radiotherapy (EBRT)

Nodular LMD is an increasingly recognized phenomenon after surgical resection with or without SRS for brain metastases. This often develops at or adjacent to the resection cavity. A small retrospective review showed salvage SRS has been used and may improve survival. This retrospective review looked at 29 patients who developed LMD after postoperative SRS and found that salvage SRS led to a median OS of 25 months compared to 5 months for salvage WBRT (*p* = 0.04). Another multi-institutional retrospective study of 147 patients found that salvage SRS for nodular LMD is associated with a higher risk of a second LMD recurrence compared to salvage WBRT (68% vs. 40%, *p* = 0.02); however, there was not a survival difference between these groups [15]. Therefore, it may be reasonable to consider salvage SRS for limited LMD while reserving WBRT for extensive intracranial disease or subsequent recurrences. To our knowledge, there is a paucity of data available regarding the use of SRS for LMD that was not associated with prior surgery or SRS for brain metastases. Focal radiotherapy such as SRS is also a consideration for patients who have CSF obstruction from focal LMD [39,52].

Other indications for the use of focal radiotherapy such as fractionated EBRT are less clear. Potential applications would be for limited spinal disease or malignant cord compression. Fractionated radiotherapy is preferred given the dose limitations of the nearby spinal cord that would limit the utility of SBRT techniques [53]. The use of fractionated EBRT for spinal lesions is not well described in the prior literature, but guidelines such as those of NCCN and EANO suggest this as an option [41]. One retrospective review reported that, of 14 patients receiving focal spine radiation, the most common treatment regimen was 30 Gy in 10 fractions [54]. For malignant cord compression, palliative focal radiation can alleviate neurologic symptoms if given soon after the onset of symptoms.

### 2.4. Craniospinal Irradiation

CSI targets the entire craniospinal axis using an external beam radiotherapy technique. CSI is an important cornerstone of treatment for some primary brain tumors including medulloblastoma as well as in the setting of LMD. Given the propensity of LMD to disseminate throughout the entire central nervous system, CSI is a potential treatment option for select patients with favorable characteristics. CSI can be used for prophylaxis in the setting of limited LMD at the time of diagnosis or in the palliative setting to improve symptoms or provide temporary disease control [55,56]. Given the large volume treated when using CSI, it is often associated with significant acute side effects including nausea, vomiting, and lymphopenia. Therefore, advanced conformal techniques including proton radiotherapy and its tissue-sparing properties have been of interest in this setting.

To date, CSI continues to be most commonly performed with photon radiotherapy. With advancements in treatment planning, photon techniques for CSI have markedly evolved over time. CSI was historically performed using three-dimensional conformal radiation therapy (3DCRT) [57]. This included the use of opposed lateral beams to treat the brain with typically 1–2 posterior fields to treat the spinal axis. This was technically challenging, with frequent difficulties with patient alignment and dose heterogeneity. Dose hotspots were of concern at the junction of the cranial and upper spinal fields with underdosing common between the upper and lower spinal fields leading to irregular dose distribution [58]. Matching fields and subsequent dose feathering to match field junctions throughout treatment increased the complexity. In recent years, more conformal techniques such as intensity modulated radiation therapy (IMRT) and volumetric modulated arc therapy (VMAT) have been adopted due to improvements in dose distribution, reduced dose to certain organs at risk, and plan robustness [59]. There is also a suggestion through modeling that the risk of secondary malignancy is lower with modern techniques due to decreased tissue exposure [60]. However, despite the use of modern photon techniques, CSI is associated with significant acute side effects, such as such as nausea, vomiting, and lymphopenia [61]. One study found that patients experienced myelosuppression (69%), dysphagia (56%), mucositis (44%), and nausea (19%) with photon CSI23. These adverse effects can have a significant impact on quality of life and can lead to treatment discontinuation. Toxicities from CSI such as myelosuppression can also impact a patient’s ability to resume systemic therapy, which may affect prognosis. However, CSI techniques are still advancing and showing promise. At least one institution utilized VMAT with a vertebral body-sparing technique for patients with LMD and found toxicity profiles to be favorable and comparable to proton CSI [62]. This may offer a feasible alternative for institutions where proton CSI is not available.

Proton radiotherapy is a type of particle radiation therapy that has a more predictable dose distribution. It is known to reduce radiation exposure to adjacent and distant organs more than traditional photon radiation (Figure 2). Protons have a number of specific properties that have therapeutic advantages. Protons are charged particles that allow for a characteristic depth–dose curve for energy deposition called the Bragg curve [63]. This occurs in part due to protons’ ability to slow down continuously as a function of depth. As these particles lose velocity, their rate of energy loss or linear energy transfer (LET) is greater. The point of highest dose deposition is called the Bragg peak, and a dose beyond this is minimal as the protons stop abruptly. Complex calculations and modeling of protons take advantage of this and allow for the precise treatment of intended targets with a negligible exit dose. The use of proton radiotherapy for malignant radiosensitive tumors has steadily increased over time [64].

Proton radiotherapy is being increasingly utilized in the United States particularly in pediatric malignancies [65,66]. Proton CSI has many theoretical benefits [67]. Now, it has become an accepted standard for any pediatric brain tumor that requires CSI, such as medulloblastoma [65]. In addition, prior studies with adult medulloblastoma patients show that proton CSI is safe and can provide a significant reduction in acute effects such as hematologic and gastrointestinal toxicities. One such study by Brown et al., which directly compared proton CSI to photon CSI patients, found that proton CSI had significantly lower rates of nausea requiring anti-emetics (26% vs. 71%), esophagitis requiring medical management (5% vs. 57%), and lymphopenia (48% vs. 65%) compared to those treated with photon CSI [68]. Concerns exist surrounding the cost of proton treatments; however, a recent exploratory cost analysis of proton versus photon CSI for adult medulloblastoma showed no significant differences in cost between the two modalities [69].

To date, there is only one randomized prospective study that has directly compared proton CSI to photon radiotherapy for LMD. Yang et al. performed a phase II trial that found both a progression-free survival (PFS) benefit to proton CSI (7.5 months vs. 2.3 months) as well as an OS benefit (9.9 vs. 6.0 months) over the use of photon involved-field radiotherapy (IFRT) [70]. In this study, IFRT was defined as WBRT or focal spine radiotherapy targeting symptomatic lesions. There were no differences in acute grade 3 or 4 toxicities noted in this trial. It should be noted that the median survival across all patients was noted to be longer than many prior reports (6.0 months versus 2–4 months historically), which may be due to selection bias. Although these results are promising and are already informing practice at some institutions, further validation with phase III prospective data is needed such as NRG BN014, an approved phase 3 randomized clinical trial of proton CSI versus IFRT for patients with breast cancer or non-small cell lung cancer LMD.

### 2.5. Chemotherapy and Targeted Therapy

Effective systemic treatment options for LMD are highly dependent on primary tumor type, the presence of targetable mutations, and CNS penetration capabilities. The most common histologies associated with LMD are non-small cell lung cancer (NSCLC) and breast cancer, which both have a wide array of potential systemic therapy options.

For NSCLC, one-third of patients harbor an EGFR mutation in which osimertinib (anti-EGFR tyrosine kinase inhibitor (TKI)) can have effective CNS penetration including for leptomeningeal metastases. In addition, an EGFR mutation in NSCLC appears to be a risk factor for developing LMD compared to wild-type NSCLC [71]. A prior study showed improved PFS for relapsed EGFR-mutant NSCLC with LMD with the use of osimertinib at the time of relapse [72]. In this study, patients had to have LMD and progression after previous EGFR-directed TKI therapy. The objective response rate of osimertinib on this study was at least 41%. Other anti-EGFR agents have shown activity in LMD such as afatinib, erlotinib, and gefitinib [73,74,75]. In a subsequent pooled analysis of 243 patients with LMD related to EGFR-mutant NSCLC, receiving EGFR-targeted TKIs, the median PFS in patients receiving any EGFR-TKI was 9.1 months and the median OS was 14.5 months. In this study, osimertinib demonstrated significantly prolonged PFS and OS in the subset of patients who were previously treated with EGFR-TKIs [76]. Anaplastic lymphoma kinase (ALK) mutations are found in less than 10% of NSCLC. The dual ALK and ROS1 inhibitor lorlatinib has been associated with durable CNS responses up to 22 months. Responses to the ALK inhibitor alectinib have also been reported [77,78].

In breast cancer, on the other hand, a major consideration is hormone receptor status. Patients with HER-2-positive breast cancer have a higher rate of CNS metastases [79]. Trastuzumab was the first HER-2 antibody drug that demonstrated a marked survival benefit for women with HER-2-positive breast cancer, with a 33% reduction in the risk of death [80]. There are many anti-HER-2 agents that have high CNS penetrance and tolerable toxicity such as trastuzumab deruxtecan [81]. There is evidence from the ROSET-BM study that demonstrated that fam-trastuzumab-deruxtecan (T-DXd) treatment was associated with favorable PFS and potentially an OS benefit for breast cancer patients with LMD [82]. In this study, the patients with LMD had a 12-month PFS and OS of 60.7% and 87.1%, respectively. For refractory heavily pretreated HER-2 breast cancer with LMD, Neratinib, a TKI, appears to be safe and effective [83]. Median OS in this study was 8 months, with a median PFS of 3.5 months. For patients without targetable mutations, cytotoxic chemotherapy regimens can be used although typically are much less effective than targeted therapy. Common single-agent drugs with known efficacy for LMD in breast cancer include high-dose methotrexate and capecitabine [84].

Intrathecal chemotherapy (ITC) has been used to treat LMD previously for many decades with modest efficacy [85]. The most commonly used ITC agents are methotrexate, thiotepa, and cytarabine. A prior randomized prospective trial found no survival benefit to intrathecal chemotherapy compared to systemic therapy for breast cancer patients and was associated with increased rates of neurotoxicity [86]. Patients on this study were randomized to receive ITC or non-intrathecal therapy, and OS was not significantly different between the two groups (18.3 vs. 30.3 weeks, *p* = 0.32). Additionally, a retrospective study of patients with LMD who received systemic therapy and radiation with or without intrathecal chemotherapy saw no difference in median OS (4 months in each group) [87]. While efficacy of intrathecal (IT) therapy with conventional chemotherapy is limited, there may be a role for IT immunotherapy or IT targeted therapy. As an example, in a study of 43 patients with LMD related to melanoma treated with IT interleukin-2 (IL-2), median OS was 7.8 months [88]. Another study identified that the expression of lipocalin-2 (LCN2) is critical for the survival of tumor cells within CSF [89]. LCN2 allows tumor cells to outcompete macrophages for iron, thereby surviving in the harsh, nutrient-poor CSF environment. This has led to the development of the IT iron chelator deferoxamine for the treatment of LMD, subject of an ongoing clinical trial, NCT05184816. Based on current evidence, the use of IT therapy for LMD can only be justified for ultra-select patients who have good performance status, controlled systemic disease, and no CNS-penetrant systemic therapy options or in the context of a clinical trial.

### 2.6. Immunotherapy

The emergence of immunotherapy options for the treatment of various malignancies has steadily increased over the past decade. Investigations involving the CNS penetrance of immunotherapy including ICI have been of particular interest in the setting of CNS metastases. It was previously believed that the CNS was “immunologically privileged” and separated from the rest of the body due to the blood–brain barrier (BBB) and lack of lymphatic drainage [90]. Recent work has indicated that this is not the case and that the immune system can bypass the BBB if a robust immune response is generated [91].

Programmed cell death 1 (PD-1), programmed cell death ligand 1 (PD-L1), and cytotoxic lymphocyte associated protein 4 (CTLA-4) inhibitors have been shown to have effective intracranial activity for certain solid tumors such as non-small cell lung cancer and melanoma brain metastases. Nivolumab and ipilimumab alone or in combination provide a progression-free survival benefit in melanoma [92]. Dual ICI therapy has also been shown to be beneficial; prior studies demonstrated a survival benefit over chemotherapy alone for stage IV NSCLC with its use [13]. The combined approach appears to hold true for advanced melanoma, as ipilimumab plus nivolumab provide the longest progression-free and OS compared with ICI monotherapy [14]. Dual ICI therapy is associated with significant toxicities, with the rate of grade 3 or 4 events occurring in 55% of patients receiving nivolumab and ipilimumab, most commonly diarrhea, rash, elevated aminotransferase levels, and colitis. A limited brain response is also achieved for brain metastases arising from NSCLC with the use of pembrolizumab [93]. These, along with numerous other novel agents, are subject to active investigations particularly for brain metastases.

The specific role for immunotherapy for LMD is not well defined. Two recent phase II studies looking at pembrolizumab for patients with LMD found it to be safe and potentially beneficial [94,95]. A single-arm phase II study using pembrolizumab for LMD from solid tumors showed a 40% rate of grade 3 or higher toxicities, without a detriment to survival (OS at 3 months 60%) [96]. Another phase II study assessing pembrolizumab in this population found rates of grade 3 or higher toxicities to be 15% with a CNS response rate of 38% in a group of 13 patients [94]. The majority of patients in both of these studies had LMD from breast cancer (85% and 62%, respectively). There is also strong data on the use of combination ICI therapy. One study looked at the use of combination nivolumab and ipilimumab in the setting of LMD, in which 18 patients were included, and overall was found to be safe [95], although it should be noted that one-third of patients experienced a grade 3 or higher toxicity on this study. These toxicities included fatigue, nausea, rash, and fever, which are commonly seen with the use of immunotherapy agents.

### 2.7. Combination Therapy—Radiation Therapy + Immune Checkpoint INHIBITORS

More recently, there has been interest in combination therapy in the treatment of metastatic cancer including those with LMD. One area gaining interest is the use of radiotherapy combined with ICI therapies. The general principles behind the effectiveness of combination radiotherapy and ICI therapies are enhanced antigen-presenting cell (APC) activation and T cell priming by radiation and reverse immune exhaustion after chronic T cell activation by PD-1/PD-L1 inhibitors [97]. In addition, there is also a phenomenon called the abscopal effect, defined as tumor growth inhibition outside the field of treatment, that may be induced with combination ICI and radiation therapy that has been documented previously. Prior studies provide evidence that combining hypofractionated radiotherapy along with ICI therapies such as anti-CTLA-4 antibodies may induce an abscopal effect [98]. One study demonstrated an abscopal-like effect when combining single-fraction SRS with ipilimumab, with greater responses seen with SRS before or concurrently with ipilimumab [99]. In this study, 20% of patients had grade 3 or 4 toxicities with the majority occurring when SRS was given within one month of a dose of ipilimumab. These adverse events included diarrhea, rash, fatigue, seizures, and CNS bleeding. It is important to note that these prior studies used focal radiotherapy with ablative doses. It is hypothesized that treating large volumes with more conventional doses of radiation may negatively affect APCs and other immunogenic cells. For example, the HN004 trial, which assessed the addition of durvalumab to radiotherapy for head and neck squamous cell carcinomas, found significantly worse survival and PFS compared to standard-of-care treatment for cisplatin-ineligible patients [100]. It is unclear if other large treatment volumes such as with CSI may also lead to a decreased effectiveness when combined with ICI therapy.

Data are limited on the use of combination therapy for LMD; however, they can be extrapolated from the brain metastases’ literature. A meta-analysis looked at the use of SRS and ICI for the treatment of brain metastases and found that the concurrent administration of these two modalities was safe, as well as potentially more effective than sequential therapy [101]. In this meta-analysis by Lehrer et al., a 1-year local control with combination ICI and SRS was 89.2% versus 67.8% for non-concurrent therapy. There was no noted difference in the rates of radionecrosis in the combination ICI and SRS population (5.3% across all studies). Other studies provide further support for the safety and efficacy of combination immunotherapy with SRS for a range of histologies of brain metastases [102,103]. A small retrospective case series of two patients with LMD from melanoma showed that the addition of concurrent proton CSI to ICI therapy after failing ICI monotherapy was well tolerated and appeared to have favorable outcomes [104].

## 3. Ongoing Trials

According to clinicaltrials.gov, there are currently six trials enrolling LMD patients receiving radiotherapy as a part of treatment (Table 1). Of these, four are investigating the use of combination therapy with targeted agents or immunotherapy in combination with radiotherapy. One trial open in Denmark is assessing proton CSI for LMD in a phase II trial (NCT05746754). As a follow-up to the recent randomized phase II trial, a phase III prospective trial, NRG BN014, comparing proton CSI versus IFRT, has recently been approved by NRG Oncology.

## 4. Future Directions

While the prognosis of LMD has historically been grim, recent advancements and ongoing research have demonstrated promising results. New therapies aim to improve dismal rates of control and survival in this population through the development of more effective or intensified therapies. Novel targeted therapies and immunotherapies are continuously emerging and can be used alone or in conjunction with other treatments to potentially improve outcomes. In particular, preliminary studies on the combination of radiotherapy with immunotherapy suggest that it is safe and may offer some evidence of improved outcomes but warrants further investigation. Specifically, focal radiation using SRS combined with immunotherapy would be of interest given the demonstrated intracranial activity of immunotherapy in solid tumors.

Currently, there are a wide array of treatment options for patients with LMD, each associated with specific toxicities that can impact quality of life. Given that survival for this group could be variable, it is important to stratify patients by prognosis. Patients with a better prognosis may benefit from more intensive therapies or, conversely, would be affected by sub-acute or chronic toxicities associated with these treatments. Patients with poorer prognoses may benefit the most from efficacious treatments with minimal acute toxicity. Currently, there are no available guidelines to help guide patient selection. Validated metrics to predict prognosis with which to guide treatment decisions are needed.

There is prospective data on the benefits of proton radiotherapy for CSI in treating LMD. Yang et al. provided phase II evidence of a survival benefit to proton CSI for LMD from breast cancer and NSCLC with acceptable toxicity profiles [52]. Further validation within more prospective randomized studies with larger cohorts is commencing. If definitive studies confirm the efficacy of proton CSI in this population, it may lead to larger-scale practice changes in the management of LMD. Patient selection metrics are lacking and continue to be important for identifying optimal candidates for aggressive therapies such as CSI.

## 5. Conclusions

Leptomeningeal disease is a devastating development in some patients with metastatic cancer. While LMD is associated with a poor prognosis, there are a number of treatment options available or are under active investigation. Radiotherapy and systemic therapies continue to evolve and improve over time with regards to the treatment of LMD. At present, validated tools to stratify patients are not available, making optimal treatment selection challenging. Tailoring the best treatment for a patient with LMD requires the careful consideration of multiple factors: tumor histology, targetable mutations, extent of disease, performance status, access to therapies, and more. A subset of patients with LMD may have more favorable prognoses and could benefit from more aggressive treatment, as well as be vulnerable to sub-acute or late toxicities due to longer survival. Improving patient selection and predictors for prognosis is crucial for identifying cohorts that would be best suited for aggressive treatments. Additional validation for newer therapies such as proton CSI and combination CSI and ICI therapies with larger randomized prospective studies is anticipated.

## Figures and Tables

**Figure 1 biomedicines-12-01792-f001:**
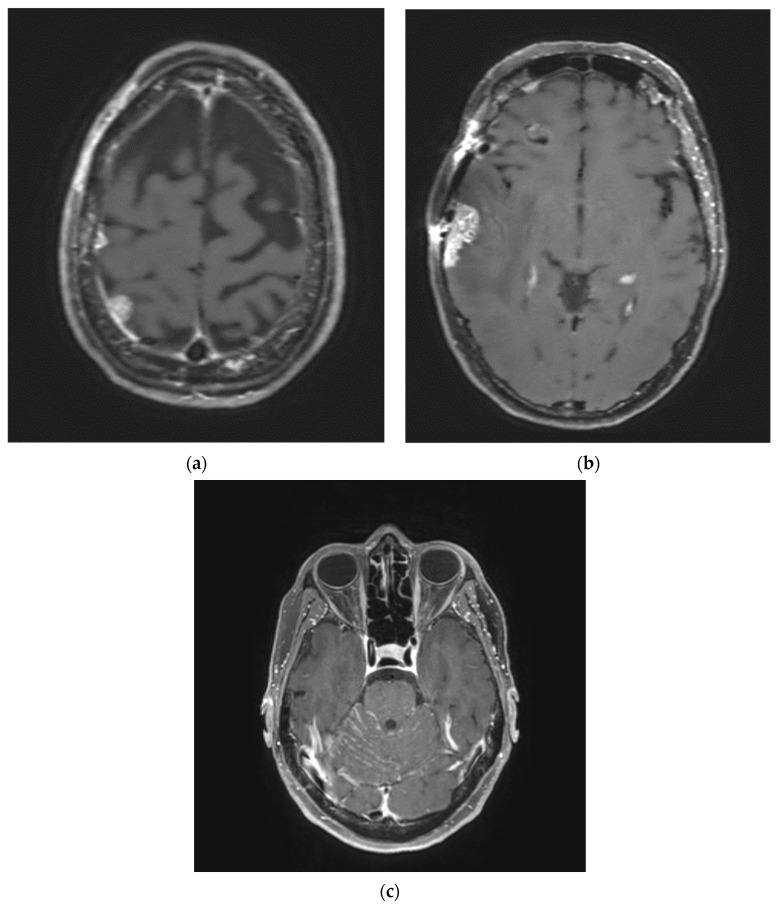
Leptomeningeal disease sub-types. (**a**) Multi-focal nodular leptomeningeal disease in the right parietal lobe; (**b**) nodular leptomeningeal disease in the right frontal lobe; (**c**) diffuse leptomeningeal disease.

**Figure 2 biomedicines-12-01792-f002:**
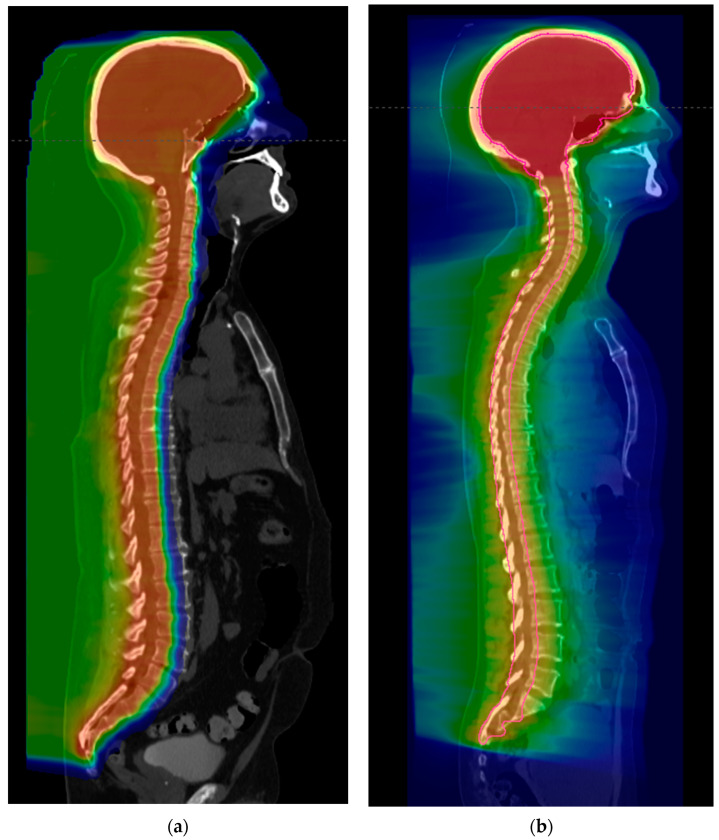
Photon versus proton craniospinal irradiation techniques. (**a**) Proton craniospinal irradiation with color wash dose distribution; (**b**) photon craniospinal irradiation using VMAT with color wash dose distribution.

**Table 1 biomedicines-12-01792-t001:** Active clinical trials for leptomeningeal disease treatment that include radiotherapy.

Study	Status	Phase	Country	Histology Specific	Drug	RT Type	Dose/ Fractions	Primary Outcome
NCT05305885	Recruiting	I	China	No	Pemetrexed	IFRT	40 Gy in 20 fx	Clinical response, treatment-related adverse events
NCT05746754	Recruiting	II	Denmark	No	None	Proton CSI	30 Gy in 10 fx	Local control
NCT04343573	Active, not recruiting	II	USA	NSCLC, Breast	None	Proton CSI, IFRT	30 Gy in 10 fx	CNS progression-free survival
NCT04192981	Recruiting	I	USA	PIK3CA mutated solid tumors	GDC-0084	WBRT	30 Gy in 10 fx	Maximum tolerated dose
NCT03719768	Active, not recruiting	I	USA	No	Avelumab	WBRT	30 Gy in 10 fx	Safety and dose-limiting toxicity
NCT04588545	Recruiting	I and II	USA	Breast	Pertuzumab, Trastuzumab	WBRT, focal brain or spine RT	30 Gy in 10 fx or 20 Gy in 5 fx	Maximum tolerated dose, overall survival

## Data Availability

No new data were created or analyzed in this study.

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
