# Peer review of "Radiotherapy and Systemic Treatment for Leptomeningeal Disease"

_biomedicines, 2024, doi:10.3390/biomedicines12081792_

Round 1

Reviewer 1 Report

Comments and Suggestions for Authors

A well-structured article, deserved to be published.

I surprised “the current work on the topic is not mentioned in the references”

More pics having deeper details can be added, if available.

Comments on the Quality of English Language

Minor revision required.

Author Response

Comments 1: A well-structured article, deserved to be published.

Response 1: Thank you for taking the time to review our manuscript and we very much appreciate your feedback. We are delighted that you agree with our work and that it deserves to be published.

Comments 2: I surprised “the current work on the topic is not mentioned in the references”

Response 2: Thank you for the comment and review. We would be happy to adjust the references but would need some additional clarification from the reviewer.

Comments 3: More pics having deeper details can be added, if available.

Response 3: Thank you for this comment. We have included three figures in the paper, including multiple pictures. Figure 1 features three pictures that represent the different sub-types of leptomeningeal disease and their classic presentations. Our second figure depicts craniospinal irradiation and offers pictures comparing the dosimetric differences between photon and proton CSI. Our third figure is a table, offering a comprehensive list of the active clinical trials on LMD that include radiotherapy.

Reviewer 2 Report

Comments and Suggestions for Authors

The manuscript “Radiotherapy and Systemic Treatment for Leptomeningeal Disease” by Frechette et al. presents a detailed overview of the evidence for the detection, prognostication, and treatment of leptomeningeal disease (LMD). In the introduction, LMD is characterized by incidence and recognition. Prognostic and optimal treatment options are also presented. The main part of the manuscript concerns treatment options for LMD, from radiotherapy, proton therapy, and immunotherapy to combination therapy. The table collects ongoing clinical trials for LMD. Future directions and conclusions showed that additional validation for newer therapies such as proton CSI, combination CSI and ICI therapies with larger randomized prospective studies is anticipated.

The Reviewer found one significant shortcoming that must be improved: the type of manuscript Review. There is no sign of mandatory steps for Systematic Review. 

Author Response

Comments 1: The manuscript “Radiotherapy and Systemic Treatment for Leptomeningeal Disease” by Frechette et al. presents a detailed overview of the evidence for the detection, prognostication, and treatment of leptomeningeal disease (LMD). In the introduction, LMD is characterized by incidence and recognition. Prognostic and optimal treatment options are also presented. The main part of the manuscript concerns treatment options for LMD, from radiotherapy, proton therapy, and immunotherapy to combination therapy. The table collects ongoing clinical trials for LMD. Future directions and conclusions showed that additional validation for newer therapies such as proton CSI, combination CSI and ICI therapies with larger randomized prospective studies is anticipated.

Response 1: Thank you for your very comprehensive and accurate summary of our paper. We appreciate your time and efforts in reviewing our work.

Comments 2: The Reviewer found one significant shortcoming that must be improved: the type of manuscript Review. There is no sign of mandatory steps for Systematic Review. 

Response 2: Thank you for the comment. After further discussion, we completely agree with your assertion and we have been working with Dr. Yajun Li to get the type of article changed to a “Review”. Our manuscript has been reworded to reflect this.

Reviewer 3 Report

Comments and Suggestions for Authors

This article  provides a comprehensive overview of the current state of radiotherapy and systemic treatments for leptomeningeal disease (LMD). The abstract effectively summarizes the scope of the review, highlighting the increasing incidence of LMD due to advancements in systemic therapies and detection methods.

I would suggest some minor corrections.

1)Please add description of the literature search methodology and clearer explanation of inclusion and exclusion criteria

2) Further discuss on the potential side effects and quality of life impacts of novel therapies in discussion.

Author Response

Comments 1: This article  provides a comprehensive overview of the current state of radiotherapy and systemic treatments for leptomeningeal disease (LMD). The abstract effectively summarizes the scope of the review, highlighting the increasing incidence of LMD due to advancements in systemic therapies and detection methods.

Response 1: Thank you for taking the time to review our work in great detail, and we appreciate the succinct summary that you have provided of our work.

Comments 2: 1)Please add description of the literature search methodology and clearer explanation of inclusion and exclusion criteria

Response 2: Thank you for the comment. We agree with your suggestion and have added a section titled “Methods” in introduction that explains our methodology for our literature search, including key terms as well as exclusion criteria.

Comments 3: 2) Further discuss on the potential side effects and quality of life impacts of novel therapies in discussion.

Response 3: Thank you for this comment and for your thorough review of our paper. Side effects of these therapies are an important piece to highlight and we agree that expanding on this would be warranted. Please find our updated manuscript where we discuss the side effects of immune checkpoint inhibitors, as well as combination ICI and radiotherapy toxicities in more detail.